# Meta-Analysis on the Effect of Contingency Management for Patients with Both Psychotic Disorders and Substance Use Disorders

**DOI:** 10.3390/jcm10040616

**Published:** 2021-02-06

**Authors:** Marianne Destoop, Lise Docx, Manuel Morrens, Geert Dom

**Affiliations:** 1Faculty of Medicine and Health Sciences, Collaborative Antwerp Psychiatric Research Institute (CAPRI), University of Antwerp, 2610 Antwerp, Belgium; Lise.docx@multiversum.broedersvanliefde.be (L.D.); manuel.morrens@uantwerpen.be (M.M.); geert.dom@uantwerpen.be (G.D.); 2Multiversum, Provinciesteenweg 408, 2530 Boechout, Belgium; 3University Psychiatric Hospital Antwerp, Campus Duffel, 2570 Antwerp, Belgium

**Keywords:** contingency management, substance use disorder, psychosis

## Abstract

Background: Substance use disorders (SUD) are highly prevalent among psychotic patients and are associated with poorer clinical and functional outcomes. Effective interventions for this clinical population are scarce and challenging. Contingency management (CM) is one of the most evidence-based treatments for SUD’s, however, a meta-analysis of the effect of CM in patients with a dual diagnosis of psychotic disorder and SUD has not been performed. Methods: We searched PubMed and PsycINFO databases up to December 2020. Results: Five controlled trials involving 892 patients were included. CM is effective on abstinence rates, measured by the number of self-reported days of using after intervention (95% CI −0.98 to −0.06) and by the number of negative breath or urine samples after intervention (OR 2.13; 95% CI 0.97 to 4.69) and follow-up (OR 1.47; 95% CI 1.04 to 2.08). Conclusions: Our meta-analysis shows a potential effect of CM on abstinence for patients with SUD and (severe) psychotic disorders, although the number of studies is limited. Additional longitudinal studies are needed to confirm the sustained effectivity of CM and give support for a larger clinical implementation of CM within services targeting these vulnerable co-morbid patients.

## 1. Introduction

Patients with a psychotic disorder and co-morbid substance use disorder (SUD) are an especially vulnerable group of psychiatric patients. Research shows that these patients have more symptoms of psychosis and depressive symptoms [1,2], higher relapse and (re)hospitalization rates [3,4], are more likely to have legal and aggression problems [4,5,6], and have higher suicide rates, as well as a lower average age at death [4,7] than psychotic patients without SUD. Of importance, these negative outcomes seem to be for the most part driven by the (continuing) substance use. Thus, targeting substance use is an essential component of the treatment of these patients. Indeed, research suggests that around 40% of psychotic patients have a lifetime SUD diagnosis [8], making the need for effective treatment strategies even more urgent [8]. Nevertheless, to date, evidence for any (effective) pharmacological or psychosocial treatment is scarce and generally of low quality [9].

Contingency management (CM), i.e., systematically rewarding behavioral change as a part of the treatment, has shown promising results in addiction research [10,11]. Increasingly, CM is also being used within the broader field of behavioral medicine. A recent meta-analysis showed an improvement in increasing physical activity and medication adherence in patients with chronic health conditions [12]. CM in the field of addiction treatment, generally entails the monetary reward of obtaining abstinence, measured with biological samples (urine, breath). Basic principles of CM are that the reward is made contingent on behavioral change, rewards are progressive (i.e., larger rewards can be earned for longer periods of successive abstinence) and the period between the behavioral change and the reward is made as short as possible [13]. Several meta-analyses have shown its effectiveness for alcohol, tobacco, amphetamine, cocaine, cannabis, opiates and poly-substance dependency, with effect sizes ranging from d = 0.31 for tobacco to d = 0.65 for opiate use [14,15,16,17,18]. Studies that compare CM with other psychosocial interventions generally conclude that CM is superior to other treatments [19]. Thus, CM is thought of as one of the most evidence-based treatments for SUDs, although its long-term effects are still debated [20].

To date, a meta-analysis on the effect of CM in patients with a dual diagnosis of psychotic disorder and SUD has not been performed. A recent Cochrane review did incorporate the effect of CM in patients with severe mental illness (psychosis, in addition to affective and personality disorders) and substance misuse. The authors could not find any differences on treatment retention, number of positive urine samples or hospitalizations between CM and standard care [21]. However, this meta-analysis did not focus on psychotic disorders specifically, obscuring interpretation for these types of patient and a large trial on CM in psychotic patients has been published after the data collection of the mentioned Cochrane review. Thus, in this paper, we aim to investigate the evidence for CM in psychotic patients with SUD by performing a meta-analysis on the available research data.

## 2. Materials and Methods

### 2.1. Search Strategy and Selection Criteria

The review was carried out in accordance with PRISMA guidelines [22]. A PubMed search was conducted for papers on contingency management and psychosis up to December 2020 using the following search terms: Schizophrenia OR Schizophrenic OR Schizo-affective OR Psychosis OR Psychotic OR “Ultra High Risk” OR “At Risk Mental State” or ARMS OR “dual disorder” OR dual-diagnosis OR “serious mental illness” OR “severe mental illness” OR SMI AND “Contingency Management”.

Eligible papers were extracted from PubMed, PsycINFO, Scopus and Web of Science databases using the following inclusion criteria: (1) English language articles published in peer-reviewed journals, (2) Tested one or more CM intervention(s) in a controlled trial aimed at substance use reduction or abstinence (3) Included patients with psychotic disorders (including schizophrenia, schizo-affective disorder, psychotic disorder not otherwise specified, schizophrenia spectrum disorder as defined by authors, ultra-high risk and at risk mental state patients). Exclusion criteria were (1) non-English papers, (2) studies not measuring the intervention of CM in substance use or abstinence, (3) non-controlled designs, (4) studies only including patients with non-psychotic disorders.

Both cross-sectional and longitudinal studies were included. In case of sample overlap between studies (as reported by the authors), only the largest one was included in order to avoid double counting.

An overview of the inclusion process can be found in Figure 1. Briefly, the above PubMed search yielded 439 results; abstract screening led to the exclusion of 203 papers, leaving 48 papers. Of these, 5 papers were included according to the above mentioned in- and exclusion criteria.

All included studies were assessed on their quality in order to account for study quality in the meta-analysis. The initial assessment of risk of bias was performed by one researcher (M.D.), and subsequently and independently by a second researcher (L.D.) by using the revised risk-of-bias tool (RoB2) [23]. The level of risk of bias is noted in Table 1.

### 2.2. Outcome Measures 

Basic demographic information (age, gender distribution), clinical variables (diagnosis, psychosis symptom severity) and effectiveness of CM were extracted for each study. Authors were contacted for additional information if data could not be extracted from the paper. 

The effectiveness of CM was calculated by (1) standardized means and standard deviations for each individual study by self-reported number of days of drug, alcohol or nicotine use in the previous 30 days and by (2) the proportion of biochemically verified negative samples. These substance use measures were primary outcome variables.

Since CM could not only be effective for decreasing substance use, but can also be useful for enhancing retention in treatment, the numbers lost to treatment and lost to follow-up were also calculated. Lost to treatment and lost to follow-up numbers were secondary outcome variables. 

If substance use in a study was presented by medians and interquartile range, means and standard deviations were estimated by the method of Wan [24].

### 2.3. Data Analysis 

Meta-analyses and subgroup analyses were carried out using Review Manager (RevMan) version 5.4 (The Cochrane Collaboration, 2020, London, UK). Data were entered into a Mantel–Haenszel analysis comparing the efficacy of CM with control across all drugs. All meta-analyses were carried out as random effects analyses due to the wide variety of CM interventions included [25]. To allow comparisons of CM to control, some multi-arm trials were collapsed into a two-arm design by averaging the effects across the treatment conditions. For example, one study [26] had four conditions (CM with either bupropion or placebo and non-contingent reinforcement with either bupropion or placebo), so the two CM conditions were collapsed together, as were the two non-contingent reinforcement conditions. The I^2^ statistic was used to assess the percentage of variability in treatment effect estimates attributable to between-study heterogeneity.

## 3. Results

### 3.1. Study Selection

In total, five studies [26,27,28,29,30] were deemed eligible and were included in the meta-analysis (total number of patients *n* = 892; patients in CM intervention group: *n* = 458; patients in control intervention group: *n* = 434; see Table 2). Five studies were randomized controlled trials (RCTs) with an intervention period between 22 days and 3 months. Four studies provided a follow-up between 6 and 18 months. All patients were outpatients.

One study focused on the use of cannabis [30], two studies investigated cigarette smoking [26,29], one studied stimulant use [27] and one mainly focused on alcohol use [28]. Abstinence was verified with diverging time intervals, from twice a day [29] to once weekly [30]. Two studies provided a fixed reward [26,29], in contrast to variable rewards [27,28,30]. 

### 3.2. Demographic

The mean age of all study groups varied between 42 and 49 years except for the groups in Rains et al. [30] who investigated CM in early psychosis. Participants were mainly male (from 65 to 100%) which is in accordance with the prevalence and incidence rates of schizophrenia in the general population [31]. 

Two studies investigated CM in severe mental illness (SMI) and in these studies only 30 and 35% met the criteria for schizophrenia or schizo-affective disorders [27,28]. Other participants were diagnosed with bipolar disorders or recurrent major depressive disorders. CM outcomes for the psychotic subgroups were not available. In most studies, symptom severity was measured at baseline by the positive and negative syndrome scale (PANSS) with a mean positive- and negative-PANSS score between 13 and 14.

### 3.3. Effectivity of CM

#### 3.3.1. Self-Reported Substance Use

After intervention, patients in the CM groups reported a significantly lower number of days using the targeted substance compared to patients in the standard care groups (SMD = −0.52, 95% CI −0.98 to −0.06; participants *n* = 806; studies *n* = 3; Z = 2.22, *p* = 0.03) (see Figure 2). The CM and control interventions lasted 3 months in each of the three studies. At follow-up, the number of days using was comparable for CM patients and for patients in the control groups (SMD = −0.36, 95% CI −0.77 to 0.05; participants *n* = 737; studies *n* = 3; Z = 1.73, *p* = 0.08) (see Figure 3). In the study by McDonell [27,28], participants were followed for 6 months, and for 18 months in the study by Rains [30].

When verifying the inter-study consistency, significant result heterogeneity was present with I^2^ scores of 79 and 86%. This suggests that disparate results are moderated by study-specific characteristics.

Since McDonell [27,28] measured the number of days using for the last 30 days after intervention and at follow-up, while Rains [30] asked the number of days using for the previous 84 and 168 days, means and SD for the study by Rains were divided by 2.8 and 5.6, respectively. The included studies measured the number of days using systematically by the Timeline Followback (TLFB) and the Addiction Severity Index (ASI).

In both nicotine studies the number of cigarettes per day was asked at baseline, and only Tidey [26] also reported this number after intervention. Patients receiving 4 weeks CM smoked significantly fewer cigarettes per day than patients in the non-contingent reinforcement group.

#### 3.3.2. Biochemically Verified Abstinence

The trend towards abstinence in CM patients can be confirmed by negative breath or urine testing. The number of participants with negative samples for the targeted substance after CM intervention was higher in CM patients compared to patients in the standard care groups, although this was not significant (CM 48%, control 38%; OR 2.13, 95% CI 0.97 to 4.69; participants *n* = 643; studies *n* = 4; Z = 1.88, *p* = 0.06) (see Figure 4). After follow-up the number of participants with negative samples was significantly higher in CM patients compared to patients in the standard care groups (CM 43%, control 35%; OR 1.47, 95% CI 1.04 to 2.08; participants *n* = 549; studies *n* = 4; Z = 2.16 *p* = 0.03) (see Figure 5).

Subgroup analyses by product were not possible since all four studies examined different substance use, i.e., amphetamine, alcohol, cannabis, and nicotine. 

The study by Tidey [26] could not be included in these analyses since only urinary cotinine levels and breath carbon monoxide levels were reported.

#### 3.3.3. Lost to Treatment and Lost to Follow-Up

No clear differences were found for “lost to treatment” between those assigned to the CM groups and participants in the standard care groups (CM 42%, control 37%; RR 1.15, 95% CI 0.90 to 1.45; participants *n* = 858; studies *n* = 4; Z = 1.12, *p* = 0.26) (Figure 6). 

However, CM participants were more likely to fall out during the follow-up period (CM 30%, control 22%; RR 1.36, 95% CI 1.04 to 1.78; participants *n* = 626; studies *n* = 3; Z = 2.26, *p* = 0.02) (Figure 7). 

Medenblik et al. [29] only reported the total number of individuals who completed the study and who were followed-up by participants, i.e., 27 and 23 of 34.

## 4. Discussion

This study aimed to investigate the effectiveness of CM in psychotic patients with SUD. Research on this topic, however, is very scarce with only five controlled trials having been reported, which is in contrast with the large number of studies and evidence on CM in non-psychotic patients with SUD [32,33]. 

The results of this meta-analysis suggest a small, significant, clinical advantage of CM on abstinence rates. Relative to those assigned to the control conditions, patients who received CM were more likely to be abstinent. On treatment retention, however, no clinical advantage was established. Results of the different studies are mixed for which several explanations could be proposed. 

Firstly, diagnostic heterogeneity between the different studies poses a challenge towards unambiguous interpretation of the results. All studies, expect for Rains’ [30], only included patients who were dependent on the targeted substance. In Rains’ study [30], 77% of the population suffered from cannabis dependence at the time of inclusion. With the inclusion criterion requiring the participants to have used cannabis at least once in 12 out of 24 weeks, it is probable that a number of patients with recreational use of cannabis were included. In general, CM studies showed poorer outcomes for individuals testing positive for targeted substances at the beginning of CM and for more heavily dependent users [34]. It is probable that those with dependence may find it harder to change their behavior compared to those with less-severe problematic use. On the other hand, CM studies suggest that CM may be particularly beneficial in those with more severe clinical presentation, i.e., for patients with two or more previous treatment episodes [34]. Rains [30], however, recruited patients from early intervention psychosis services with limited treatment history. Taken together, the partly dependent study population with limited treatment history can explain the negative results of CM in the study by Rains, in contrast to the other studies including older patients with substance dependence [30]. 

In regard to the diagnosis of psychosis, study populations differ as well. Whereas all studies included stable, treatment-seeking outpatients with mild psychotic symptoms, Rains [30] only focused on early psychosis patients, while McDonell [27,28] included patients with SMI, i.e., schizoaffective disorder, bipolar disorder and major depressive disorder. Since the two studies of McDonell [27,28] are prominent in the results of this meta-analysis, the positive effect of CM for patients with SUD and psychosis must be interpreted carefully. It might be possible that patients with co-morbid SUD and SMI (with psychosis as a subgroup) benefit from CM, while patients with SUD and psychosis (non-SMI) do not respond to CM. Larger studies with homogeneous well described study populations are needed to confirm the preliminary positive effect of CM. Weinstock, Alessi, and Petry [35] demonstrated that patients achieve improved outcomes in CM relative to standard care, regardless of the psychiatric symptom severity. Moreover, they demonstrated that, while patients with higher levels of psychiatric symptom severity exhibit poorer retention in standard treatment, retention was improved for patients with greater symptom severity in CM. Due to the small number of studies included in this meta-analysis, subgroup analyses or meta-regression analyses on the relation between psychotic symptom severity and effectivity of CM were not possible. 

Most co-morbidity studies on CM investigated patients with PTSD, major depressive disorders and antisocial personality disorders, which makes it difficult to translate the relationship between psychiatric symptoms and the effectivity of CM for psychotic patients with SUD. Actually, this current meta-analysis is the first review article on CM in patients with SUD and psychosis and points to the important limitation of the heterogeneity of study populations. Therefore, more studies with homogeneous psychotic patients with SUD are needed to more thoroughly investigate the effectivity of CM. 

Secondly, CM and control interventions differed substantially. Both fishbowl and voucher-based CM intervention methods were used, with variable reward magnitudes and both escalating and fixed reward schedules. Although fishbowl, as well as voucher based CM interventions have been found to be effective in substance abuse research [14,36], the efficacy of CM has also been shown to increase with more frequent reinforcement opportunities, immediacy of reinforcement, higher perceived reinforcement value (although the actual value may be low), escalating reinforcement schedules and resetting of reinforcement after failing to meet the reinforcement criterion [37]. A minimal length of 8 to 12 weeks of CM intervention has also been named to be of key importance in its efficacy [13,37]. Furthermore, it has been hypothesized that when it comes to efficacy, there is an interaction between characteristics of the study population and CM intervention (e.g., a higher magnitude of CM is needed in patients with poorer prognosis) [34,38]. The sparse number of studies so far hampers firm recommendations about what form of CM intervention is most effective in dually diagnosed psychotic patients, further highlighting the importance of more thorough research on this topic. 

Nevertheless, this meta-analysis does find a positive effect of CM compared to treatment as usual (TAU) in dually diagnosed psychotic patients, which is in line with the widely studied effect of substance abuse treatment [39]. Hypothetically, the efficacy of CM interventions in psychotic patients with SUD is based on altered reward related decision making. CM is based on the principles of operant conditioning: the reinforcement of target behavior leads to an increase in this behavior. This is also the principle on which cognitive behavioral programs used in addiction treatment are based, but CM might have the advantage of being more tailored to the elevated delay discounting rates in substance use and psychotic disorders [40,41,42]. Delay discounting refers to the process in which the subjective value of a reward diminishes as a function of time. Alterations in delay discounting causes patients to be drawn to smaller, immediate rewards over larger, postponed rewards. The immediacy of reinforcement in CM, therefore, might be a more suitable alternative for the immediate reinforcing effects of substance use, than the often long(er) term benefits of a sober life style [43]. An alternative—or complementary—hypothesis can be found in the described deficits in the reward system’s wanting, i.e., the motivating effect of the anticipatory pleasure before a reward is “consummated”. Both substance use and psychotic disorders have been linked to motivational anhedonia, while consummatory pleasure (“liking*”*) appears to be more intact [44,45]. The high frequency of reinforcement in CM might thus help to compensate for the motivational deficits of patients, increasing the chance of behavioral change. Therefore, several studies also experimented with CM-interventions rewarding other behaviors [46].

An important question remains on how long the effect of a CM intervention is sustained with regard to substance use. The results of this meta-analysis suggest that the effect of CM on abstinence lasts for 6 months [27,28]. In the study by Rains [30], patients were followed for 18 months and the results from this study were negative. While some studies in substance abusing populations show a relatively long-term effect, a large number of others suggest that the effect is relatively short term [37]. Consistent with these latter findings, improvements in health behaviors in patients with chronic health conditions tend to dissipate when the contingency is removed [12]. Interventions need to be implemented to prolong the effect, i.e., by extending the period of CM reward or by adding other therapeutic interventions. As to the former, Petry et al. [47] showed a clear improvement in outcome comparing a 12 week CM program with a 6 week program. This suggests that CM might need to be considered as a long-term (continuing) intervention in supporting a patient’s abstinence. In addition, alternative strategies can be developed to enhance the effect of a CM intervention, e.g., by combining CM with other psychosocial interventions (CBR, CRA) or with pharmacotherapy [10,48]. However, findings here are not consistent, and a recent meta-analysis found no evidence that combining CM with other (behavioral) interventions improved the long-term effects of CM treatment in substance misusing populations [49]. These conflicting findings raise the question whether CM should be considered a “primary” intervention or whether CM should be considered as an important add-on on top of other behavioral interventions to increase the effectiveness. In support of the latter, is a recent study showing that adding CM to a behavioral intervention significantly increases the effectiveness [50]. Of importance, adding CM to other behavioral interventions did not only increased the effectivity on smoking cessation but also recently proved to be cost-effective [50].

The current study results should be interpreted in the light of several limitations. First, we identified a considerable amount of diagnostic heterogeneity between study populations. Severity of the SUD and psychotic diagnoses differed significantly between the studies. Moreover, it must be mentioned that only the use of cannabis, nicotine, alcohol, and amphetamines were investigated. Studies on the effect of CM in psychotic patients using depressant and other types of drugs are lacking. Secondly, CM and control interventions varied substantially. The lack of uniform outcome measures used throughout the different studies (substance use related biological and self-report, treatment retention, etc.), limits comparison between studies and warrants caution in the interpretation of the results. This is a characteristic of many studies in mental health when evaluating complex psychosocial interventions, highlighting the need to develop internationally accepted standard outcome variables, allowing for better comparison among future studies. Thirdly, the number of studies included in this meta-analysis is very low, which makes it hard to draw clear conclusions on the effectivity of CM in patients with psychotic disorders and SUD. Future studies with homogeneous dual diagnosis populations are urgently needed, in order to support the effectiveness of CM in psychotic patients with SUD. Fourthly, the outcome measures lost to treatment and lost to follow-up are measures of attrition, but are not the sole indices of retention. They do not indicate survival (time until attrition). Furthermore, time-in-treatment was not evaluated, and CM has been found to have positive effect on the length of time-in-treatment [51,52]. Finally, we included only English-language research articles published in peer-reviewed journals. This might have increased the level of bias in our study results, because we did not include foreign language studies, unpublished studies, partially published studies, and studies in “grey” literature sources.

## 5. Conclusions

Taken together, in addition to the mounting evidence of CM’s effectivity in the treatment of SUD-patients and changing broader health-related behaviors, our meta-analysis provides, albeit based on a limited number of studies, preliminary evidence for its effectivity within patients with severe SUD and psychiatric co-morbidity (i.e., psychotic disorders). Indeed, implementing CM might be specifically important within the management of patients who would otherwise a poorer prognosis in standard care [34]. However, despite the growing evidence of its effectivity, CM is—as of yet—still underutilized in our care systems and many barriers remain [10,39,53]. As to the latter, ethical and political discussions, i.e., whether rewarding patients for abstinence is an appropriate strategy, often over-shadow the evidence on CM’s (cost) effectiveness [10,53]. Thus, additional studies in the future, allowing for a larger meta-analytic approach, are needed to confirm the effectivity of this approach within patients with severe psychiatric co-morbidities and provide additional support for larger clinical implementation within services targeting these vulnerable co-morbid patients.

## Figures and Tables

**Figure 1 jcm-10-00616-f001:**
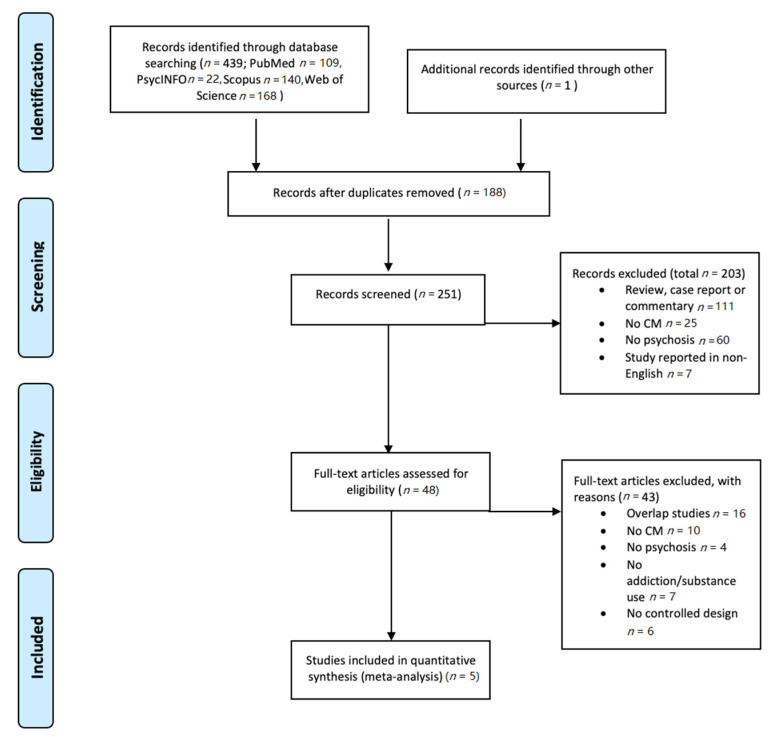
PRISMA flowchart of the selection of studies. CM: contingency management.

**Figure 2 jcm-10-00616-f002:**
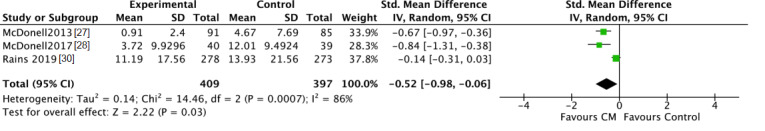
Forest plot of the comparison CM versus standard care after intervention, number of days using.

**Figure 3 jcm-10-00616-f003:**
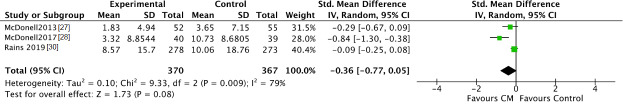
Forest plot of the comparison CM versus standard care after follow-up, number of days using.

**Figure 4 jcm-10-00616-f004:**
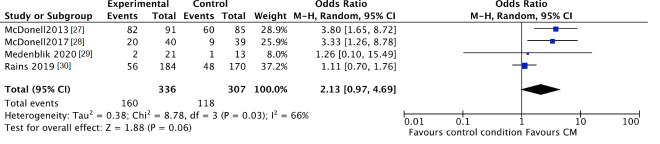
Forest plot of the comparison CM versus standard care after intervention, percentage of participants with negative samples.

**Figure 5 jcm-10-00616-f005:**
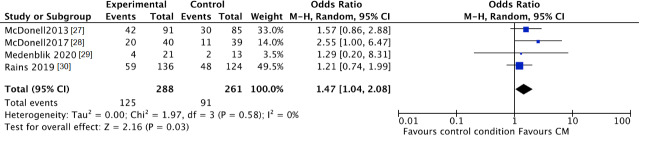
Forest plot of the comparison CM versus standard care after follow-up, percentage of participants with negative samples.

**Figure 6 jcm-10-00616-f006:**
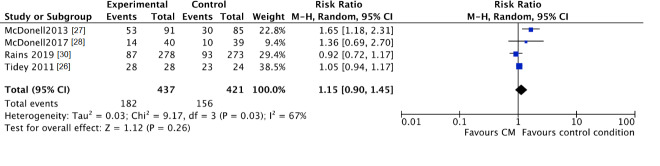
Forest plot of the comparison CM versus standard care after intervention, percentage of participants lost to treatment.

**Figure 7 jcm-10-00616-f007:**
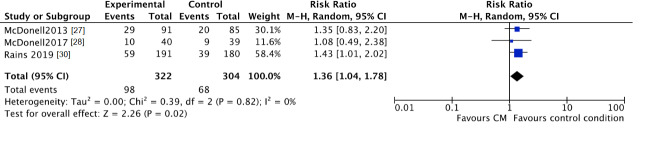
Forest plot of the comparison CM versus standard care after intervention, percentage of participants lost to follow-up.

**Table 1 jcm-10-00616-t001:** The level of risk of bias by the revised risk-of-bias tool (RoB2).

Author	Randomisation	Intervention Bias	Missing Outcome Data	Measurement	Reporting Bias
McDonell 2013 [27]	Low risk	Low risk	Low risk	Low risk	Low risk
McDonell 2017 [28]	Low risk	Low risk	Low risk	Low risk	Low risk
Medenblik 2020 [29]	Some concerns	Low risk	Some concerns	Low risk	Some concerns
Rains 2019 [30]	Low risk	High risk	High risk	Low risk	Low risk
Tidey 2011 [26]	Low risk	Low risk	Low risk	Low risk	Low risk

**Table 2 jcm-10-00616-t002:** Overview of studies.

Authors, Design	Sample Inclusion Criteria	Sample Exclusion Criteria	CM Intervention	Reward	Control Intervention	Follow-Up	Results
McDonell, 2013, RCT [27]	Have used stimulants during past 30 d, methamphetamine/amphetamine/cocaine dependence and schizophrenia/schizoaffective disorder/bipolar I or II/recurrent major depressive disorder	Organic brain disorder, dementia or medical disorders or psychiatric symptoms severe enough to compromise safe participation	CM: 3 m 3 x/w breath and urine samples, 1 opportunity for each negative sample and 1 additional for each week of continuous stimulant abstinence + TAU *n* = 91	Prize draws from container of tokens respresenting magnitudes of reinforcement (50% “good job” and 50% tangible prize	3 m Noncontingent reinforcement for each urine sample regardless result + TAU: psychoeducation + mental health care by case manager + psychiatric medication management + group treatment + housing and vocational services *n* = 85	3 m	CM group less likely to complete treatment period, more likely to submit a stimulant-negative urine test. CM group lower levels of alcohol use, injection drug use, fewer psychiatric symptoms, less likely to be admitted for psychiatric hospitalization.
McDonell, 2017, RCT [28]	Alcohol dependence and schizophrenia/schizoaffective disorder/bipolar I or II/recurrent major depressive disorder, alcohol use on >4d of the last 30d, enrollment in outpatient addiction group treatment	Comorbid drug dependence, medical or psychiatric severity that would compromise safe participation	CM: 12 w 3 x/w urine samples, 3 prize draws for each EtG-negative urine sample and 1 additional prize draw for each week of continuous alcohol abstinence + TAU *n* = 40	Prize draws from container of tokens respresenting magnitudes of reinforcement (50% “good job” and 50% tangible prize	12 w Noncontingent reinforcement for each urine sample regardless of EtG result + TAU: psychoeducation + mental health care by case manager + psychiatric medication management + group treatment + housing and vocational services *n* = 39	3 m	CM group had longest duration of alcohol abstinence, fewer days of drinking, fewer heavy drinking episodes and lower EtG levels. They were more likely to submit stimulant-negative urine and smoking-negative breath samples.
Medenblik, 2020, RCT pilot [29]	Smoked for at least 1 year, 18–70 y, at least 10 cig/d, sufficient English, willing to make a smoking cessation attempt, criteria for schizophrenia/schizoaffective disorder/psychotic disorder	History of AMI last 6 m, contraindication for NRT, unwilling to quit other forms of nicotine, pregnancy, criteria for current mania, current incarceration of inpatient hospitalization	iCOMMIT: 30 d behavioral therapy in the form of mCM + pharmacotherapy for smoking cessation + five sessions of cognitive-behavioral smoking cessation couseling *n* = 21	Money checks for bioverification of abstinence through using phone app	ITC: 30 d pharmacotherapy for smoking cessation + five sessions of cognitive-behavioral smoking cessation couseling *n* = 13	6 m	No statistically significant difference between iCOMMIT and ITC groups.
Rains, 2019, RCT [30]	On an EIP service caseload, THC once in 12/24 w, 18–36 y, living in stable accommodation, sufficient English	Compulsory treatment or court	12 w, weekly CM sessions with immediate reward with vouchers if negative urinalaysis for THC *n* = 278	Variable reward schedule with £ 5 for every 2 clean samples	12 w optimised TAU psychoeducational intervention recommended in EIP practice *n* = 273	18 m	No statistically significant difference in cannabis use, in engagement in work of education and positive psychotic symptoms. CM significant improvement in time to acute psychiatric admission
Tidey, 2011, RCT [26]	Schizophrenia/schizoaffective disorder, >18y, >20 cig/d, >6 FTND, stable on medication for >2 m, interested in quitting smoking	Pregnancy, positive breath alcohol level or urine drug toxicity test, medical condition contraindicating bupropion, very high psychiatric symptom severity	CM + BUP: 3w 3 x/w urine and CO samples + bupropion 150mg 3d and bupropion 2 × 150 mg 4–22 d *n* = 12 CM + PLA: 3 w 3 x/w urine and CO samples + placebo *n* = 16	USD 25 for attending study sessions and increased by USD 5 for each abstinent sample	NRT + BUP: 3 w 25 dollar for attending study sessions and providing urine samples at each visit + bupropion 150 mg 3 d and bupropion 2 × 150 mg 4–22 d *n* = 11 NRT+ PLA: 3w USD 25 for attending study sessions and providing urine samples at each visit + placebo *n* = 13	No	Significantly decreased cotinine and CO levels in CM group.

RCT: Randomized Controlled Trial; CC: Case Control Study; CM: Contingency Management; TAU: Treatment as Usual; NRT: Nicotin Replacement Therapy; EIP: Early Intervention Psychosis, FTND: Fagerström Test of Nicotine Dependence; BUP: Bupropion; PLA: placebo; CO: carbon monoxide.

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
