# Peer review of "Meta-Analysis on the Effect of Contingency Management for Patients with Both Psychotic Disorders and Substance Use Disorders"

_jcm, 2021, doi:10.3390/jcm10040616_

Round 1

Reviewer 1 Report

This manuscript describes a meta-analysis of studies that assessed the effect of contingency management (CM) for patients with both psychotic disorders and substance use disorders. The manuscript is very well written and easy to read. However, I encountered a number of issues that significantly dampened my enthusiasm for the manuscript since. I will briefly expand on my points of more substantial concern.

My main concern is the small number (5) and the heterogeneity of papers included in the meta-analysis. Both conditions negatively impact the conclusions that may be reached. Heterogeneity refers to the MC protocols included (voucher-based and fishbowl), the disparity of the substances (amphetamine, alcohol, cannabis and nicotine), the outcome measures and the different follow-up periods in which the treatment effect is analyzed. In my opinion, this heterogeneity in the included studies this heterogeneity preclude to conduct the meta-analysis under good conditions.

Besides this main concern, there a several problems that should be addressed.

Maybe I have missed something, but I cannot find any term related to substance use among the terms used to perform the search.

Did the authors use any exclusion criteria for the selection of the articles?

Why did the authors not include outcomes measures such as abstinence rates or continued abstinence, which are common in clinical studies to evaluate the effectiveness of treatments?

Related, in my opinion, the lost of follow-up is not a valid variable to evaluate the effectiveness of the treatment. On the other hand, the authors should specify how the lost to treatment has been evaluated and defined.

The studies included in the meta-analysis have very dispersed follow-up periods, from 3 to 18 months, and it seems that the results on the effectiveness of CM have been collapsed into a single period, as if all the studies had the same or similar follow-up.
this can provide misleading data and conclusions.

In my opinion, these methodological limitations make the conclusions of the study do not have a high impact in the literature on this important issue.

Author Response

Response to Reviewer 1 Comments

Point 1: Contingency management (CM) has been found to have one of the largest effect sizes among psychosocial treatments for substance use disorders. Although CM is based on the operant conditioning principle of positive reinforcement as a means of increasing the frequency and likelihood of a behavior, what remains unclear is whether CM’s effect size differs across various patient populations being treatment for substance use disorder (SUD). Using meta-analysis of the extant literature on the application of CM to treat patients with SUD and serious mental illness, the authors evaluate whether CM’s beneficial effect on SUD applies to patients with co-morbid psychotic disorders and SUD. Given the co-morbidity of SUD and SMI, this is an important question regarding the generalizability of CM’s effects across SUD clinical subpopulations. The manuscript is well-organized.

Response 1: We thank the reviewer for this positive feedback.

Point 2: However, the manuscript does have several flaws to be addressed. The manuscript notes that “a meta-analysis of the effect of CM in patients with a dual diagnosis of psychotic disorder and SUD has not been performed.” That is true insofar as there does not appear to be a paper that whose review of the literature on applications of CM to patients with co-morbid SUD and psychosis included a meta-analysis of treatment effects. However, McPherson and colleagues (2018) did conduct an extensive literature review (that included 31 published CM studies) that examined CM outcomes among under-served patients including patients with SUD and SMI (including psychosis). That literature review (McPherson et al., Substance Abuse and Rehabilitation, 2018, 9, 43–57) included at least two studies that appear to be germane to the present manuscript’s efforts (Bellack et al., 2006 & Ries et al. 2004). Indeed, the study by Bellack and colleagues (2006) found a beneficial effect of CM on both abstinence and treatment retention. The study by Ries and colleagues (2004) also found a beneficial effect of CM on abstinence.

Two additional literature reviews by Hjorthoj et al. (2009) and Horsfall et al. (2009) identified two studies by Sigmon and colleagues (2006 & 2000) as well as the aforementioned study by Bellack et al. (2006), all of which included patients with co-morbid SUD and psychosis. Furthermore, the studies by Sigmon and colleagues found a beneficial CM effect on abstinence.

The exclusion of these studies from this manuscript suggests flaws in the search strategy presented in the manuscript. The search strategy presented in section 2.1 of the manuscript did not include the following key terms: “serious mental illness,” “severe mental illness,” “SMI,” “contingent,” “reinforcement,” “incentive,” “reward,” “dual-diagnosis,” “dual disorder.” The inclusion of these terms would likely have captured the studies by Bellack et al., Sigmon et al., and Ries et al..

Response 2: We thank the reviewer for the additional literature reviews of McPherson (2018), Hjorthoj et al. (2009) and Horsfall et al. (2009) and integrated these studies in the manuscript.

The scope of this paper is to investigate the effectivity of contingency management in patients with psychotic disorders and SUD. As noticed by the first reviewer, it might be possible that the current search strategy did not capture all studies with patients with SUD and SMI (possibly including a subgroup of psychotic patients). However, a recent cochrane meta-analysis on the effect of CM on patients with SMI and SUD was performed by Hunt (2019) and included only 4 papers as well, which we thoroughly studied of course for inclusion in the current meta-analysis. Of the meta-analysis of Hunt only the two papers of McDonell (2013 & 2017) were included in the current meta-analysis since psychotic diagnosis were described as a subgroup (see response 3).

To meet the concerns of the first reviewer, we broadened the search strategy as suggested, by the following key terms: Schizophrenia OR Schizophrenic OR Schizo-affective OR Psychosis OR Psychotic OR "Ultra High Risk" OR "At Risk Mental State" or ARMS OR dual-diagnosis OR "dual disorder" OR "serious mental illness" OR "severe mental illness" OR SMI AND "Contingency Management". Based on the suggestion of the second reviewer we used Web of science and Scopus as well to ensure no study was missed. As expected, the extended search did not yield additional papers for inclusion (see adapted figure 1).

Our intention is to measure the effectivity of CM in patients with comorbidity of psychosis and SUD. Hence, we included only studies that tested CM interventions in a controlled trial aimed at substance use reduction or abstinence. The studies of Sigmon (2000 & 2006), Bellack (2006) and Ries (2004) were captured by the search strategy, but were excluded since they were not designed to test the effectivity of CM. Bellack (2006) and Ries (2004) evaluated treatment programs, but did not measure effectivity of CM an sich. Sigmon (2000 & 2006) did not use a case-control designs.

Concerning the other suggested key terms (contingent, reinforcement, incentive, or reward), a quick literature search was performed and demonstrated a scope being too broad. Since we specifically want to focus on the effectivity of CM and CM is a widely used and accepted concept, we are confident the current search strategy captures all papers we need to investigate the effectivity of CM in patients with comorbidity of psychosis and SUD.

Point 3: Another flaw in this meta-analysis is that two of the five included studies (both by McDonell et al. in 2013 and 2017) did not evaluate CM for patients with SUD and psychosis, exclusively. In addition to patients with co-morbid SUD and psychosis, the studies by McDonell and colleagues also included patients with other SMIs, i.e. bipolar and major depressive disorders. Therefore, the effect sizes from these two studies are not exclusive to patients with co-morbid SUD and psychosis. Furthermore, the two studies by McDonell and colleagues represent 66% (2/3) of the studies that the meta-analysis used to assess CM’s effect on self-reported abstinence (presented in Figures 2 & 3), 50% (2/4) of the studies used to assess CM’s effect on biochemically-verified abstinence (presented in Figures 4 & 5), 50% (2/4) of the studies used to assess treatment attrition, and 66% (2/3) of the studies used to assess loss to follow-up. Given the prominence of the McDonell studies in this meta-analysis, the manuscript appears to be evaluating the effect of CM for patients with co-morbid SUD and SMI (not psychosis exclusively). Indeed, the manuscript alludes to this limitation by noting the significant result heterogeneity. Therefore, presenting the meta-analysis as an evaluation of CM for patients with co-morbid SUD and psychosis might be misleading.

Response 3: We totally agree with this important comment and emphasized this limitation already several times in the first version. To underline the importance of this limitation even more, we rewrote this paragraph in the discussion. Furthermore, we added a limitation section and presented the conclusions even more careful.

Point 4 The manuscript limits analyses of CM effect on retention to treatment attrition and loss to follow-up. Because that approach does not evaluate a CM effect for time (survival) in treatment, one can only speculate as to whether CM patients remained in care longer than non-CM patients despite have attrition rates comparable to non-CM patients. This limitation should be presented in the manuscript.

Response 4: Loss to follow-up and treatment retention are one of the used outcome measures in some of the studies included. As the reviewer rightly indicates this is not a direct effectiveness variables. However, treatment retention can be considered an important proxy for treatment effectivity. Indeed, specific in the case of patients with psychotic disorders, treatment retention and continuity has been associated in different studies with improved outcomes (both on the symptomatic level as on the functional level) (Chang, 2015). Overall, an important limitation remains the heterogeneity of the outcome measures, used in the studies included in this meta-analysis (se also our response to Point 5 of reviewer 3). We highlighted this in our limitation section:

Text change:

“The lack of uniform outcome measures used throughout the different studies (substance use related biological and self-report, treatment retention, etc. ), limits comparison between studies and warrants caution in the interpretation of the results. This is a characteristic of many studies in mental health when evaluating complex psychosocial interventions, highlighting the need to develop internationally accepted standard outcome variables, allowing for better comparison among future studies.

Point 5 The following copyedits also are needed:

Line 22: Change “show” to “shows”

Line 30: Insert “symptoms of psychosis” after “positive”

Line 36: Move citation [8] to after “diagnosis”

Line 38: Change “general” to “generally”

Line 40: Insert a comma after “treatment”

Line 52: Change “Is” to “is”

Line 68: Delete the underscore after “[22].”

Line 95: Add “1” after “Table”

Line 127: Change “1” to “2”

Line 127: Delete “an” and change “RCT” to “RCTs”

Line 129: Chane “outpatient” to “outpatients”

Line 130: Change “Table 1.” To “Table 2.”

Line 139: Change the commas to periods.

Line 142: Add a citation to substantiate the prevalence and incidence data to which the manuscript refers.

Line 152 & Line 177: Change “their main product” to “the targeted substance”

Line 197: Change “Figure 5” to “Figure 6”

Line 200: Change “Figure 6” to “Figure 7”

Line 222: Change “criterium” to “criterion”

Line 223: Delete the apostrophe and add a comma after “weeks”

Line 265: Delete “to make”

Line 279: Change “suited” to “suitable”

Line 280: Change “positive” to “reinforcing”

Line 280: Change “clean” to “sober” (the latter is stigmatizing language)

Line 283: Place the period inside the closing quote mark after “consummated”

Line 291: Change “of” to “by”

Response 5: All copyedits are inserted.

REFERENCES CITED IN THIS REVIEW

Bellack, AS, Bennett, ME, Gearon, JS, Brown, CH, & Yang, Y. (2006). A randomized clinical trial of a new behavioral intervention for drug abuse in people with severe and persistent mental illness. Archives of General Psychiatry, 63, 426−432.

Chang WC, Chan GH, Jim OT, Lau ES, Hui CL, Chan SK, Lee EH, Chen EY. Optimal duration of an early intervention programme for first-episode psychosis: randomised controlled trial. Br J Psychiatry. 2015 Jun;206(6):492-500.

Hjorthoj C, Fohlmann A, Nordentoft M. (2009). Treatment of cannabis use disorders in people with schizophrenia spectrum disorders – A systematic review. Addictive Behaviors, 34, 846–851.

Horsfall J, Cleary M, Hunt GE, Walter G. (2009). Psychosocial treatments for people with co-occurring severe mental illnesses and substance use disorders (dual diagnosis): A review of empirical evidence. Harvard Review of Psychiatry, 17(1), 24–34.

Ries RK, Dyck DG, Short R, Srebnik D, Fisher A, Comtois KA. (2004). Outcomes of managing disability benefits among patients with substance dependence and severe mental illness. Psychiatr Serv., 55(4), 445–447.

Sigmon SC, Higgins ST. (2006). Voucher-based contingent reinforcement of marijuana abstinence among individuals with serious mental illness. J Subst Abuse Treat., 30, 291–5.

Sigmon SC, Steingard S, Badger GJ, Anthony SL, Higgins ST. (2000). Contingent reinforcement of marijuana abstinence among individuals with serious mental illness: a feasibility study. Exp Clin Psychopharmacol., 8, 509–17.

Reviewer 2 Report

Contingency management (CM) has been found to have one of the largest effect sizes among psychosocial treatments for substance use disorders. Although CM is based on the operant conditioning principle of positive reinforcement as a means of increasing the frequency and likelihood of a behavior, what remains unclear is whether CM’s effect size differs across various patient populations being treatment for substance use disorder (SUD). Using meta-analysis of the extant literature on the application of CM to treat patients with SUD and serious mental illness, the authors evaluate whether CM’s beneficial effect on SUD applies to patients with co-morbid psychotic disorders and SUD. Given the co-morbidity of SUD and SMI, this is an important question regarding the generalizability of CM’s effects across SUD clinical subpopulations. The manuscript is well-organized. However, the manuscript does have several flaws to be addressed.

The manuscript notes that “a meta-analysis of the effect of CM in patients with a dual diagnosis of psychotic disorder and SUD has not been performed.” That is true insofar as there does not appear to be a paper that whose review of the literature on applications of CM to patients with co-morbid SUD and psychosis included a meta-analysis of treatment effects. However, McPherson and colleagues (2018) did conduct an extensive literature review (that included 31 published CM studies) that examined CM outcomes among under-served patients including patients with SUD and SMI (including psychosis). That literature review (McPherson et al., Substance Abuse and Rehabilitation, 2018, 9, 43–57) included at least two studies that appear to be germane to the present manuscript’s efforts (Bellack et al., 2006 & Ries et al. 2004). Indeed, the study by Bellack and colleagues (2006) found a beneficial effect of CM on both abstinence and treatment retention. The study by Ries and colleagues (2004) also found a beneficial effect of CM on abstinence.

Two additional literature reviews by Hjorthoj et al. (2009) and Horsfall et al. (2009) identified two studies by Sigmon and colleagues (2006 & 2000) as well as the aforementioned study by Bellack et al. (2006), all of which included patients with co-morbid SUD and psychosis. Furthermore, the studies by Sigmon and colleagues found a beneficial CM effect on abstinence.

The exclusion of these studies from this manuscript suggests flaws in the search strategy presented in the manuscript. The search strategy presented in section 2.1 of the manuscript did not include the following key terms: “serious mental illness,” “severe mental illness,” “SMI,” “contingent,” “reinforcement,” “incentive,” “reward,” “dual-diagnosis,” “dual disorder.” The inclusion of these terms would likely have captured the studies by Bellack et al., Sigmon et al., and Ries et al..

Another flaw in this meta-analysis is that two of the five included studies (both by McDonell et al. in 2013 and 2017) did not evaluate CM for patients with SUD and psychosis, exclusively. In addition to patients with co-morbid SUD and psychosis, the studies by McDonell and colleagues also included patients with other SMIs, i.e. bipolar and major depressive disorders. Therefore, the effect sizes from these two studies are not exclusive to patients with co-morbid SUD and psychosis. Furthermore, the two studies by McDonell and colleagues represent 66% (2/3) of the studies that the meta-analysis used to assess CM’s effect on self-reported abstinence (presented in Figures 2 & 3), 50% (2/4) of the studies used to assess CM’s effect on biochemically-verified abstinence (presented in Figures 4 & 5), 50% (2/4) of the studies used to assess treatment attrition, and 66% (2/3) of the studies used to assess loss to follow-up. Given the prominence of the McDonell studies in this meta-analysis, the manuscript appears to be evaluating the effect of CM for patients with co-morbid SUD and SMI (not psychosis exclusively). Indeed, the manuscript alludes to this limitation by noting the significant result heterogeneity. Therefore, presenting the meta-analysis as an evaluation of CM for patients with co-morbid SUD and psychosis might be misleading.

The manuscript limits analyses of CM effect on retention to treatment attrition and loss to follow-up. Because that approach does not evaluate a CM effect for time (survival) in treatment, one can only speculate as to whether CM patients remained in care longer than non-CM patients despite have attrition rates comparable to non-CM patients. This limitation should be presented in the manuscript.

The aforementioned citations are listed at the conclusion of this review.

The following copyedits also are needed:

Line 22: Change “show” to “shows”

Line 30: Insert “symptoms of psychosis” after “positive”

Line 36: Move citation [8] to after “diagnosis”

Line 38: Change “general” to “generally”

Line 40: Insert a comma after “treatment”

Line 52: Change “Is” to “is”

Line 68: Delete the underscore after “[22].”

Line 95: Add “1” after “Table”

Line 127: Change “1” to “2”

Line 127: Delete “an” and change “RCT” to “RCTs”

Line 129: Chane “outpatient” to “outpatients”

Line 130: Change “Table 1.” To “Table 2.”

Line 139: Change the commas to periods.

Line 142: Add a citation to substantiate the prevalence and incidence data to which the manuscript refers.

Line 152 & Line 177: Change “their main product” to “the targeted substance”

Line 197: Change “Figure 5” to “Figure 6”

Line 200: Change “Figure 6” to “Figure 7”

Line 222: Change “criterium” to “criterion”

Line 223: Delete the apostrophe and add a comma after “weeks”

Line 265: Delete “to make”

Line 279: Change “suited” to “suitable”

Line 280: Change “positive” to “reinforcing”

Line 280: Change “clean” to “sober” (the latter is stigmatizing language)

Line 283: Place the period inside the closing quote mark after “consummated”

Line 291: Change “of” to “by”

REFERENCES CITED IN THIS REVIEW

Bellack, AS, Bennett, ME, Gearon, JS, Brown, CH, & Yang, Y. (2006). A randomized clinical trial of a new behavioral intervention for drug abuse in people with severe and persistent mental illness. Archives of General Psychiatry, 63, 426−432.

Hjorthoj C, Fohlmann A, Nordentoft M. (2009). Treatment of cannabis use disorders in people with schizophrenia spectrum disorders – A systematic review. Addictive Behaviors, 34, 846–851.

Horsfall J, Cleary M, Hunt GE, Walter G. (2009). Psychosocial treatments for people with co-occurring severe mental illnesses and substance use disorders (dual diagnosis): A review of empirical evidence. Harvard Review of Psychiatry, 17(1), 24–34.

Ries RK, Dyck DG, Short R, Srebnik D, Fisher A, Comtois KA. (2004). Outcomes of managing disability benefits among patients with substance dependence and severe mental illness. Psychiatr Serv., 55(4), 445–447.

Sigmon SC, Higgins ST. (2006). Voucher-based contingent reinforcement of marijuana abstinence among individuals with serious mental illness. J Subst Abuse Treat., 30, 291–5.

Sigmon SC, Steingard S, Badger GJ, Anthony SL, Higgins ST. (2000). Contingent reinforcement of marijuana abstinence among individuals with serious mental illness: a feasibility study. Exp Clin Psychopharmacol., 8, 509–17.

Author Response

Response to Reviewer 2 Comments

Point 1: I thank the editors for the opportunity to collaborate as a reviewer on the Journal of Clinical Medicine. I would also like to congratulate the authors of the manuscript "Meta-analysis on the effect of Contingency Management for patients with both Psychotic Disorders and Substance Use Disorders", for the made efforts in their study.

Response 1: We thank the reviewer for the recognition of our efforts and work.

Nevertheless, some important weaknesses of the study design display a weak validity and reliability of the presented data.

Minor modifications:

Point 2: The authors should review and modify the numbering and title of the tables.

Response 2: The numbering and titles of tables and figures are reviewed and modified.

Point 3: In Table 2 they should explain all the abbreviations.

Response 3: All abbreviations are explained in table 2 below.

Point 4: Authors should include a section indicating the limitations in their study.

Response 4: As requested, we added a limitation section in the manuscript.

Major modifications:

Point 5: The databases used are insufficient. The authors should carry out the search on the “Web of Science” and “Scopus”.

Response 4: We thank the reviewer for this relevant suggestion. We carried out the search on the “Web of Science” and “Scopus” as well. This extra search went well, but did not yield additional papers for inclusions. The results of the extended search are described under the section results and presented in Figure 1 (see also comment 2 from Reviewer 1).

Point 6: The number of articles selected for the meta-analysis are insufficient, as authors already point out. Only 4 studies are with dependent patients. More studies are needed to obtain a valid and reliable conclusion.

Response 6: We fully agree that the number of studies in this meta-analysis is scarce. We pointed out this scarcity several times in the abstract, results, discussion and conclusions of the first version of the manuscript. In the revised version we presented the results even more carefully and we stress the urgent need for additional studies for this especially vulnerable group of usually young patients, especially the lack of other effective treatment options.

We are confident that the number of studies is sufficient. Firstly, because the number of patients included in the different RCT’s is 892 in total. Secondly, because the meta-analysis is carried out appropriate with all outcomes being meaningfully pooled and results provided sufficiently similar (see guidelines Cochrane: http://cccrg.cochrane.org/sites/cccrg.cochrane.org/files/public/uploads/meta-analysis_revised_december_1st_1_2016.pdf). A comparable Cochrane meta-analysis on the effect of CM for patients with SUD and SMI, only included 4 papers in total (Hunt, 2019).

Point 7: The authors show an erroneous conclusion, since the studies employed do not use depressant drugs (heroin, methadone,...), these being the most used.

Response 7: We totally agree with the reviewer and incorporated this lack of evidence in the limitation section as well.

Reviewer 3 Report

I thank the editors for the opportunity to collaborate as a reviewer on the Journal of Clinical Medicine. I would also like to congratulate the authors of the manuscript "Meta-analysis on the effect of Contingency Management for patients with both Psychotic Disorders and Substance Use Disorders", for the made efforts in their study.
Nevertheless, some important weaknesses of the study design display a weak validity and reliability of the presented data.

Minor modifications:

The authors should review and modify the numbering and title of the tables.
In Table 2 they should explain all the abbreviations.
Authors should include a section indicating the limitations in their study.

Major modifications:

The databases used are insufficient. The authors should carry out the search on the “Web of Science” and “Scopus”.
The number of articles selected for the meta-analysis are insufficient, as authors already point out. Only 4 studies are with dependent patients. More studies are needed to obtain a valid and reliable conclusion.
The authors show an erroneous conclusion, since the studies employed do not use depressant drugs (heroin, methadone,...), these being the most used.

Author Response

Response to reviewer 3 comments

Point 1: This manuscript describes a meta-analysis of studies that assessed the effect of contingency management (CM) for patients with both psychotic disorders and substance use disorders. The manuscript is very well written and easy to read. However, I encountered a number of issues that significantly dampened my enthusiasm for the manuscript since. I will briefly expand on my points of more substantial concern.

Response 1: We thank the reviewer for this positive feedback.

Point 2: My main concern is the small number (5) and the heterogeneity of papers included in the meta-analysis. Both conditions negatively impact the conclusions that may be reached. Heterogeneity refers to the MC protocols included (voucher-based and fishbowl), the disparity of the substances (amphetamine, alcohol, cannabis and nicotine), the outcome measures and the different follow-up periods in which the treatment effect is analyzed. In my opinion, this heterogeneity in the included studies this heterogeneity preclude to conduct the meta-analysis under good conditions.

Response 2: We fully agree that the data found in our study warrant very careful interpretation. Both the heterogeneity and the limited number of studies represent a limitation of this meta-analysis. These limitations have been mentioned throughout the discussion and in the limitation section. However, we do think that, albeit this limitation, our choice of meta-analysis is defendable. Firstly, a total of 892 patients were studied in the included RCT’s and the meta-analysis is carried out appropriate with all outcomes being meaningfully pooled and results provided sufficiently similar according to Cochrane criteria (Ryan, 2016). A Cochrane meta-analysis on the effect of CM in patients with SUD and SMI only included 4 papers and faced similar heterogeneity in applied methods (Hunt et al. 2019). Secondly, heterogeneity in research methods is – unfortunately – a challenge in almost all psychiatric studies.

Hunt G.E.; Siegfried N.; Morley K.; Brooke-Sumner C.; Cleary M. Psychosocial interventions for people with both severe mental illness and substance misuse. Cochrane Database Syst Rev 2019, 12(12), CD001088.

Ryan R; Cochrane Consumers and Communication Review Group. ‘Cochrane Consumers and Communication Group: meta-analysis. http://cccrg.cochrane.org, December 2016 (accessed DATE)

Besides this main concern, there a several problems that should be addressed.

Point 3: Maybe I have missed something, but I cannot find any term related to substance use among the terms used to perform the search.

Response 3: Indeed, we did not include any term related to substance use in the search strategy. Our aim was to capture all papers on CM in patients with psychosis. This strategy yielded to a couple of papers on CM for psychotic patients with incentives for approving adherence with antipsychotic medication (Noordraven, 2014) or exercise (Thyer, 1984). These papers were excluded based on the reported inclusion criterium ‘Tested one or more CM intervention(s) in a controlled trial aimed at substance use reduction or abstinence’. In figure 1 the number of papers excluded due to lack of substance use by participants, is reported. Apparently, the great majority of CM papers focuses on substance use reduction or abstinence.

Noordraven EL, Audier CH, Staring AB, Wierdsma AI, Blanken P, van der Hoorn BE, Roijen LH, Mulder CL. Money for medication: a randomized controlled study on the effectiveness of financial incentives to improve medication adherence in patients with psychotic disorders. BMC Psychiatry. 2014 Dec 2;14:343. doi: 10.1186/s12888-014-0343-3. PMID: 25438877; PMCID: PMC4258939.

Thyer BA, Irvine S, Santa CA. Contingency management of exercise by chronic schizophrenics. Percept Mot Skills. 1984 Apr;58(2):419-25. doi: 10.2466/pms.1984.58.2.419. PMID: 6739238.

Point 4: Did the authors use any exclusion criteria for the selection of the articles?

Response 4: Based on the inclusion criteria we excluded papers reported in non-English, studies not measuring the intervention of CM at substance use/abstinence, non-controlled designs or studies only including patients with non-psychotic disorders or diagnoses not specified. An overview of the excluded papers is presented in figure 1. In the manuscript we described these exclusion criteria more explicitly in the method section and in the limitation section.

Point 5: Why did the authors not include outcomes measures such as abstinence rates or continued abstinence, which are common in clinical studies to evaluate the effectiveness of treatments?

Response 5: Indeed, in most clinical studies on addiction abstinence rates are measured. In our study we reported all the different outcome measured used in the different studies. This revealed a large heterogeneity in outcome measures being used by the different authors. In the current meta-analysis the abstinence was measured by negative breath and urine testing (see results 3.3.2 and figures 4&5). The common self-reported outcome measure in all included studies was use of substances by ASI or TLFB (number of days using the targeted substance).

We added a sentence in the limitation section :

“The lack of uniform outcome measures used throughout the different studies (substance use related biological and self-report, treatment retention, etc. ), limits comparison between studies and warrants caution in the interpretation of the results. This is a characteristic of many studies in mental health when evaluating complex psychosocial interventions, highlighting the need to develop internationally accepted standard outcome variables, allowing for better comparison among future studies.

Point 6: Related, in my opinion, the lost of follow-up is not a valid variable to evaluate the effectiveness of the treatment. On the other hand, the authors should specify how the lost to treatment has been evaluated and defined.

Response 6: This remark is indeed closely related to the previous point. Loss of follow-up and treatment retention are one of the used outcome measures in some of the studies included. As the reviewer rightly indicates this is not a direct effectiveness variable. However, treatment retention can be considered an important proxy for treatment effectivity. Indeed, specific in the case of patients with psychotic disorders, treatment retention and continuity has been associated in different studies with improved outcomes (both on the symptomatic level as on the functional level) (Chang, 2015).

Chang WC, Chan GH, Jim OT, Lau ES, Hui CL, Chan SK, Lee EH, Chen EY. Optimal duration of an early intervention programme for first-episode psychosis: randomised controlled trial. Br J Psychiatry. 2015 Jun;206(6):492-500.

Point 7: The studies included in the meta-analysis have very dispersed follow-up periods, from 3 to 18 months, and it seems that the results on the effectiveness of CM have been collapsed into a single period, as if all the studies had the same or similar follow-up.
this can provide misleading data and conclusions.

Response 7: The reviewer addresses the important and interesting issue on the long-term effect of CM. In the current meta-analysis, four studies provided a follow-up between 6 and 18 months (Medenblik 2020, Mc Donell 2013&2017 and Rains 2019), one study did not measure the effect of CM at follow-up so was not included in the follow-up results (Tidey, 2011). 3 out of 4 follow-up studies measured the effectivity of CM at 6 months follow-up. Only Rains (2019) reports a negative effect of CM at 18 months follow-up.  Therefore, we agree with the reviewer that it might be possible that the effect of CM on abstinence lasts for 6 months, but dissipates on the long-term, which is in accordance with literature on CM for individuals with chronic health conditions (Ellis, 2020). A separate paragraph about these follow-up issue was already incorporated in the first version of the manuscript.

Point 8: In my opinion, these methodological limitations make the conclusions of the study do not have a high impact in the literature on this important issue.

Response 8: We do appreciate the reviewer’s straight-forward opinion. As indicated in our different responses to the comments of the reviewer, we agree on different limitations (and highlight these in our discussion and limitation section). However, as also include in our responses we do think that the choice of a meta-analysis and the subsequent results are defendable. In addition, specifically because of the (clinical) importance of the topic we think that our study is very timely in bringing together the current data (and their limitations) on this topic, allowing future studies to build further on this body of evidence.

Round 2

Reviewer 1 Report

In my opinion, the scarcity and variety of the included studies prevent us from drawing valid conclusions

Author Response

Point 1: In my opinion, the scarcity and variety of the included studies prevent us from drawing valid conclusions

Response 1: We do appreciate the reviewer’s straight-forward opinion and totally agree on these limitations (and highlight these in our discussion and limitation section). Therefore, the limitation section is as follows:

‘The current study results should be interpreted in the light of several limitations. First, we identified a considerable amount of diagnostic heterogeneity between study populations. Severity of the SUD and psychotic diagnoses differed significantly between the studies. Moreover, it must be mentioned that only the use of cannabis, nicotine, alcohol and amphetamines were investigated. Studies on the effect of CM in psychotic patients using depressant and other types of drugs are lacking. Secondly, CM and control interventions varied substantially. The lack of uniform outcome measures used throughout the different studies (substance use related biological and self-report, treatment retention, etc. ), limits comparison between studies and warrants caution in the interpretation of the results. This is a characteristic of many studies in mental health when evaluating complex psychosocial interventions, highlighting the need to develop internationally accepted standard outcome variables, allowing for better comparison among future studies. Thirdly, the number of studies included in this meta-analysis is very scarce, which make it hard to draw clear conclusions on the effectivity of CM in patients with psychotic disorders and SUD. Future studies with homogeneous dual diagnosis population are highly needed, in order to support the effectiveness of CM in psychotic patients with SUD. Finally, we included only English-language research articles published in peer-reviewed journals. This might have increased our bias in our study results, because we did not include foreign language studies, unpublished studies, partially published studies, and studies in ‘grey’ literature sources.’

Furthermore, we are very careful with conclusions:

‘Taken together, in addition to the mounting evidence of CM’s effectivity in the treatment of SUD-patients and changing broader health-related behaviors, our meta-analysis provides, albeit based on a limited number of studies, preliminary evidence for its effectivity within patients with severe SUD and psychiatric co-morbidity (i.e. psychotic disorders).‘

However, we do think that the choice of a meta-analysis and the subsequent results are defendable. In addition, specifically because of the (clinical) importance of the topic we think that our study is very timely in bringing together the current data (and their limitations) on this topic, allowing future studies to build further on this body of evidence.

Reviewer 2 Report

Review of revised manuscript JCM-1071733

The authors have addressed this reviewer’s concerns well, though some concerns remain and should be addressed.

On line 20, the Abstract states that “CM was not effective for treatment retention…”.

This is an excessively definitive statement given the very limited number of studies included in the meta-analysis as well as the limitation of using treatment attrition and loss to follow-up as the sole indices of retention.  Although those two indices are measures of attrition, they do not indicate survival (time until attrition).  Furthermore, time in treatment was not evaluated and CM has been found to have a positive effect on time in treatment (see citations below).  The distinction between attrition and time-in-treatment should be explained further in the limitations.  For example, while CM and non-CM patients might have identical attrition rates in any given study, when the attrition occurs (i.e. the time in treatment until attrition) could differ between the two groups.

The following copyedits are needed:

Lines 51 & 52: Change “tabacco” to “tobacco”

Line 54: Change “Is” to “is”

Line 121: Change “Colaboration” to “Collaboration”

Line 135: Change “Nicotne” to “Nicotine”

Lines 167 & 236 & 300: Change “of” to “by”

Line 246: Change “homogenic” to “homogeneous”

Line 336: Change “population” to “populations”

Table 2: Change “respresenting” to “representing”, “couseling” to “counseling”, “urinalaysis” to “urinalysis”

REFERENCES CITED IN THIS REVIEW

Petry, N.M., Alessi, S.M., Carroll, K.M., Hanson, T., MacKinnon, S., Rounsaville, B., Sierra, S., 2006. Contingency management treatments: reinforcing abstinence versus adherence with goal-related activities. J. Consult. Clin. Psychol. 74, 592–601.

Petry, N.M., Barry, D., Alessi, S.M., Rounsaville, B.J., Carroll, K.M., 2012. A randomized trial adapting contingency management targets based on initial abstinence status of cocaine-dependent patients. J. Consult. Clin. Psychol. 80, 276–285.

Petry, N.M., Weinstock, J., Alessi, S.M., Lewis, M.W., Dieckhaus, K., 2010. Group-based randomized trial of contingencies for health and abstinence in HIV patients. J. Consult. Clin. Psychol. 78, 89–97.

Author Response

Point 1: The authors have addressed this reviewer’s concerns well, though some concerns remain and should be addressed.

Response 1: We thank the reviewer for this positive feedback.

Point 2: On line 20, the Abstract states that “CM was not effective for treatment retention…”.

This is an excessively definitive statement given the very limited number of studies included in the meta-analysis as well as the limitation of using treatment attrition and loss to follow-up as the sole indices of retention.  Although those two indices are measures of attrition, they do not indicate survival (time until attrition).  Furthermore, time in treatment was not evaluated and CM has been found to have a positive effect on time in treatment (see citations below).  The distinction between attrition and time-in-treatment should be explained further in the limitations.  For example, while CM and non-CM patients might have identical attrition rates in any given study, when the attrition occurs (i.e. the time in treatment until attrition) could differ between the two groups.

REFERENCES CITED IN THIS REVIEW

Petry, N.M., Alessi, S.M., Carroll, K.M., Hanson, T., MacKinnon, S., Rounsaville, B., Sierra, S., 2006. Contingency management treatments: reinforcing abstinence versus adherence with goal-related activities. J. Consult. Clin. Psychol. 74, 592–601.

Petry, N.M., Barry, D., Alessi, S.M., Rounsaville, B.J., Carroll, K.M., 2012. A randomized trial adapting contingency management targets based on initial abstinence status of cocaine-dependent patients. J. Consult. Clin. Psychol. 80, 276–285.

Petry, N.M., Weinstock, J., Alessi, S.M., Lewis, M.W., Dieckhaus, K., 2010. Group-based randomized trial of contingencies for health and abstinence in HIV patients. J. Consult. Clin. Psychol. 78, 89–97.

Response 2: We agree that the statement that ‘CM is not effective for treatment retention’ is a definitive statement. Therefore, we removed this statement in the abstract and we added this important limitation to the limitation section as follows:

Fourthly, the outcome measures lost to treatment and lost to follow-up are measures of attrition, but are not the sole indices of retention. They do not indicate survival (time until attrition). Furthermore, time-in-treatment was not evaluated and CM has been found to have positive effect on time-in-treatment [51, 52, 53].’

Point 3: The following copyedits are needed:

Lines 51 & 52: Change “tabacco” to “tobacco”

Line 54: Change “Is” to “is”

Line 121: Change “Colaboration” to “Collaboration”

Line 135: Change “Nicotne” to “Nicotine”

Lines 167 & 236 & 300: Change “of” to “by”

Line 246: Change “homogenic” to “homogeneous”

Line 336: Change “population” to “populations”

Table 2: Change “respresenting” to “representing”, “couseling” to “counseling”, “urinalaysis” to “urinalysis”

Response 3: All copyedits are inserted.

Reviewer 3 Report

The authors have made the changes correctly.

Author Response

Point 1: The authors have made the changes correctly.

Response 1: We thank the reviewer for the positive feedback.
